# Effect of Different Aluminum Substitution Rates on the Structure of Tobermorite

**DOI:** 10.3390/ma12223765

**Published:** 2019-11-15

**Authors:** Wei Liao, Wenqing Li, Zhenggang Fang, Chunhua Lu, Zhongzi Xu

**Affiliations:** State Key Laboratory of Materials-Oriented Chemical Engineering, College of Materials Science and Engineering, Nanjing Tech University, Nanjing 210009, China

**Keywords:** hydrothermal synthesis, Al-substituted tobermorite, ^29^Si MAS–NMR, ^27^Al MAS–NMR, adsorption capacity

## Abstract

Tobermorite and Al-substituted tobermorite with varying Al/(Si + Al) molar ratios were synthesized via the static hydrothermal method. The effect of different Al substitution ratios on the structure of tobermorite was investigated by XRD, SEM–EDS, TG–DSC, FT–IR, ^29^Si, and ^27^Al MAS–NMR. The interplanar crystal spacing of (002) increased with the increment of the Al substitution ratio. The maximum ratio of the Al substitution for Si in the structure of tobermorite was about 15 at %, which was limited by the compositions of tobermorite and the electrostatic repulsive forces caused by Al substitution. The DSC curve of the tobermorite with an Al/(Al + Si) molar ratio of 0.15 showed a larger exothermic peak area at 850 °C, indicating that the Al-O-Si bond was formed in the structure of tobermorite and the Al participated in the recrystallization. The ^29^Si and ^27^Al MAS–NMR results confirmed that tetrahedral Al was introduced into the structure of the tobermorite. New silicon sites, Q^2^(1Al) and Q^3^(1Al), formed by the Al substitution for Si in bridging and non-bridging sites, were detected. Compared to the tobermorite without Al substitution, the adsorption capacity of Al-substituted tobermorite with Al/(Al + Si) = 0.15 to Na^+^ was improved by 44%.

## 1. Introduction

Tobermorite, a layer calcium silicate hydrate mineral with an ideal composition of Ca_5_Si_6_H_2_O_18_·4H_2_O, is a naturally existing hydrated calcium silicate mineral [1]. In 1982, Komarneni et al. [2] reported that the tobermorite would show cation exchange and cation selectivity after the Si was partially substituted by Al. Komarneni and Roy [3] classified the Al-substituted tobermorite as a new family of selective cation exchangers in 1983. Since then, the Al-substituted tobermorite has attracted extensive attention not only in terms of selectivity, but also in terms of structure, insulation, stability, and so on.

There are three types of tobermorite, named 1.4 nm, 1.13 nm, and 0.9 nm tobermorite, according to the interplanar spacing of (002) crystal plane, respectively [4,5], and the 1.13 nm tobermorite is the most common type that can be found in nature. In numerous studies, the 1.13 nm tobermorite and its Al-substituted forms are mostly synthesized under hydrothermal conditions from various raw materials, such as fly ash [6,7], trachyte rock [8], zeolites [1], carbonated opoka [9], or CaO-SiO_2_-H_2_O systems [10]. Furthermore, Jauberthie et al. [11] reported that the silica tetrahedron sheets were formed along the ab plane initially and then grew into a layered structure along the c-axis. The structural analysis and property studies of 1.1 nm Al-substituted tobermorite have been carried out by several researchers [12,13,14,15,16,17]. The Al substitution for Si could occur in the bridging tetrahedron sites and the non-bridging tetrahedron sites [18,19]. Black et al. [20] reported that tetrahedral Al was found in 1.1 nm Al-substituted tobermorite based on the Al 2p binding energy. Furthermore, they found that the Si 2p binding energy decreased after the Al substitution for Si, which suggested one of two alternatives of charge balancing.

However, little attention has been paid to the maximum Al substitution ratio in the structure of tobermorite. In consideration of the composition of tobermorite and the regional charge imbalance after Si^4+^ was substituted by Al^3+^, the Al substitution ratio should be limited in a certain range. This charge imbalance in the substituted site will lead to the neighboring Si tetrahedrons unavailable for Al substitution. This substitution limit is one of the critical factors that should be considered to obtain pure Al-substituted tobermorite since the excess Al may lead to impurities in the synthesized samples.

The objective of this work is to get further understanding about the incorporation of Al into the structure of tobermorite and investigate the limit of Al substitution ratio for Si in the structure of tobermorite. Tobermorite and its different Al substitution form were synthesized under static hydrothermal environment. XRD, SEM–EDS, TG–DSC, FT–IR and ^29^Si and ^27^Al MAS–NMR were used to characterize the effects of different Al substitution ratio on the structure and morphology of tobermorite. The adsorption capacity of tobermorite, with and without Al substitution, to the Na^+^ was investigated with a flame photometer.

## 2. Experimental Work

### 2.1. Chemicals and Synthesis

In the present research, CaO (analytically pure, Shang Hai Ling Feng Chemical Reagent Co. Ltd., Suzhou, China), SiO_2_ (nanopowder, 99.5% metals basis, 50 nm, Aladdin, Shanghai, China) and Al(OH)_3_ (analytically pure, Shang Hai Ling Feng Chemical Reagent Co. Ltd., Suzhou, China) were used for various syntheses with a target mole ratio of Ca/(Si + Al) = 5:6, Al/(Al + Si) = 0~0.2. The water to solid ratio was fixed at 12 by mass. Each reaction slurry was prepared by stirring the starting materials with a magnetic stirrer. Then the slurry was transferred into a 40 mL PTFE lining stainless steel autoclave respectively and hydrothermally treated at 180 °C for 15 hours. After hydrothermal treatment, the solid products were washed several times with distilled water and dried at 80 °C for 24 hours.

### 2.2. Characterization

The phase composition of the solid products was characterized by X-ray diffraction (XRD, Rigaku D/Max-2500, Rigaku Ltd., Tokyo, Japan) with Cu-Kα radiation. Data were collected over the angular range of 2θ from 5° to 70°, with a step size of 0.02° and scan speed of 10°/min. A scanning electron microscopy energy dispersive spectrometer (SEM–EDS, JSM-6510, JEOL, Tokyo, Japan) was used to observe the micro-morphology and analyze the elementary composition. Simultaneous thermal analysis (TG–DSC, STA449C, Netzsch, Bavaria, Germany) was developed to observe mass loss and thermal transformation at a heating rate of 10 °C/min in N_2_ atmosphere ranging from 40 °C to 900 °C. The Fourier transform infrared spectra (FT–IR, Frontier MIR/FIR Std, Perkin Elmer, Waltham, MA, USA) of samples were acquired between 400 and 4000 cm^−1^ by suspending samples in KBr pallet. The magic angle spin nuclear magnetic resonance (^29^Si and ^27^Al MAS–NMR) study was performed at 300 K with a frequency of 79.49 MHz on a Bruker AV-400D spectrometer using acetone as the solvent.

### 2.3. Cation Adsorption Capacity

The flame photometer (FP650, AOPU Analytical instrument, Shanghai, China), an analytical instrument based on the principle of emission spectrometry, was used to investigate the structural changes of tobermorite caused by Al substitution. NaOH was used as an adsorbate. The content of Na^+^ was determined according to the characteristic spectrum and the intensity of light wave emitted by Na when it returns from the excited state to the ground state. The test was conducted as follows: 0.15 g of each sample was added to 1200 μL of NaOH solution with a concentration about 0.5 mol/L. The contact time for samples was 10 mins before the mixtures were centrifuged at 5000 rpm for 3 mins. Then 50 μL of supernatant liquor was transferred into a volumetric flask with micro sampling meter and diluted to 100 mL for the flame photometry test. The concentration of Na^+^ before and after adsorption and the adsorption capacity were calculated via Equations (1) and (2):(1)$$c({\mathrm{Na}}^{+})\;=\;\frac{2\times R({\mathrm{Na}}^{+})}{M({\mathrm{Na}}_{2}O)}\times D$$
(2)$$\mathrm{AC}\;=\;\frac{[{c(\mathrm{Na}}^{+}{)}_{\mathrm{before}}-{c(\mathrm{Na}}^{+}{)}_{\mathrm{after}}]\times V}{m}$$
where R(Na^+^) is the reading of Na on the flame photometry (g/L). M(Na_2_O) is the relative molecular mass of Na_2_O (g/mol). D, which equals to 2000, is the dilution ratio of the centrifugal clear liquid. c(Na^+^) is the calculated concentration of Na^+^ (mol/L). V, which equals to 1200 μL, is the volume of NaOH solution used for adsorption. m, which equals to 0.15 g, is the mass of sample used for adsorption and AC is the calculated adsorption capacity of sample to Na^+^ (mol/g).

## 3. Results and Discussion

### 3.1. XRD and SEM–EDS

The XRD results of the as-synthesized tobermorite and its Al substitution form with different Al/(Al + Si) molar ratios are shown in Figure 1. The interplanar crystal spacing marked in Figure 1 was calculated via Bragg equation (Equation (3)):(3)2dsinθ=nλ
where d is the interplanar crystal spacing, θ is the diffraction angle, λ is the wavelength of X-ray (Cu-Kα diffraction, λ = 0.1542 nm), and n is diffraction series (n = 1).

Well-crystallized Al-tobermorite was synthesized under static hydrothermal environment (Figure 1). The diffraction intensity increased with the Al/(Al + Si) molar ratio, indicating that the introduction of Al promoted the polymerization degree of the silicate chains of tobermorite. A small diffraction peak of Al(OH)_3_, near 2θ equals to 18°, was detected when the Al/(Al + Si) molar ratio increased to 0.15 and 0.2, indicating that the Al(OH)_3_ was produced by the excess Al. No hydrogarnet was detected even in the synthesized products with Al/(Al + Si) ratio of 0.15 and 0.2. This showed a different result obtained by Xiaoling Qu who reported that hydrogarnet was always detected from Al/(Al + Si) = 0.06 at different reaction times [4]. The diffraction peak between the 2θ range of 5–10° is corresponding to the (002) crystal plane of tobermorite. The position of this diffraction peak shifted to a lower angle range with the increasing of Al/(Al + Si) molar ratio, indicating the increment of (002) interlayer spacing. The increment of interplanar crystal spacing can be explained by the radius difference between Al^3+^ and Si^4+^. Due to the larger radius of Al^3+^ than Si^4+^, the structure of tobermorite would become looser in the z-axis direction after Al substitution for Si.

SEM-EDS was used to characterize the morphology and elementary composition of Al-tobermorite with various Al/(Al + Si) molar ratio. As shown in Figure 2, well-crystallized plate-like tobermorite with different Al/(Al + Si) molar ratio was successfully synthesized. Compared to the tobermorite (Figure 2a), the main crystal size and crystallinity of Al-substituted tobermorite were improved which consistent with the results of XRD. The elemental composition of the samples was detected by EDS and parts of the corresponding results were presented beside the SEM image. Only Ca, O, Al, and Si were taken into consideration when the relative content of each elements was calculated. As expected, no Al was detected in the sample with Al/(Al + Si) = 0 (Figure 2a). For other samples the relative content of Al increased with the initial Al/(Al + Si) molar ratio.

Figure 3 shows the correlation between the initial Al/(Al + Si) molar ratio in reaction slurries and the final Al/(Al + Si) molar ratio calculated by EDS analysis. Each value in Figure 3 is an average of several detecting points. Obviously, when the initial Al/(Al + Si) molar ratio increases from 0 to 0.15, the detected Al/(Al + Si) molar ratio showed a linear relationship with it. However, the detected Al/(Al + Si) molar ratio is no longer increase when the initial Al/(Al + Si) molar ratio is above 0.15. The detected Al/(Al + Si) molar ratio is about 0.15 even the initial Al/(Al + Si) molar ratio increased to 0.2. This trend indicates that the molar ratio of Al substitution for Si is limited to less than 0.15, which is related to the compositions of tobermorite and the regional charge imbalance after Si^4+^ was substituted by Al^3+^. This conclusion will be further confirmed and discussed in Section 3.3. Once the limitation of substitution was reached, the Al/(Al + Si) molar ratio in the structure of Al-tobermorite would be stable and the excess Al would be formed Al(OH)_3_ (Figure 1).

### 3.2. TG-DSC

The TG-DSC curves of tobermorite and Al-substituted tobermorite with Al/(Al + Si) molar ratio of 0.15 are shown in Figure 4a,b respectively. For the DSC curve in Figure 4a, there is a wide endothermic peak before 200 °C due to the loss of free water and dehydration of interlayer water. The weak endothermic peak at 680 °C indicated the Si-OH chain was cracked and dehydroxylation occurred. The structure of tobermorite was completely destroyed and turned into an unordered, metastable structure. The exothermic peak at 840 °C was corresponding to the β-wollastonite that was produced from the recrystallization of tobermorite [21,22,23]. Due to the similar structure and thermal behavior between C-S-H(I) and tobermorite, it could not be identified clearly by XRD and TG–DSC. Thus, the synthesized sample might be a mixture of 1.13 nm tobermorite and C-S-H (I). However, according to the results of XRD and TG-DSC, tobermorite remains the main component. Compared to tobermorite, the DSC curve of tobermorite with an Al/(Al + Si) molar ratio of 0.15 in Figure 4b shows a similar endothermic position and exothermic peak. However, the exothermic peak of recrystallization appeared at 850 °C, which showed a larger peak area. It indicated that the Al-O-Si bond was formed in the structure of tobermorite and the Al participated in the recrystallization [22]. Thus, more energy was needed during the recrystallization process. Furthermore, no endothermic peak of the decomposition of hydrogarnet at 300–550 °C appeared in the DSC curve of tobermorite with Al/(Al + Si) molar ratio of 0.15. Thus, it is deduced that no hydrogarnet formed in the synthesis progress which is consistent with the results of XRD (Figure 1). 

### 3.3. FT–IR

The FI–IR spectra in the range of 4000–400 cm^−1^ of the hydrothermally-synthesized Al-tobermorite with various Al/(Al + Si) molar ratios are illustrated in Figure 5. 

For all the samples, the FT–IR spectra showed a broad peak at 3446 cm^−1^ and a band at 1638 cm^−1^. These two peaks are the characteristic peak of O-H stretching and H-O-H bending vibration assigned to the bound H_2_O of Al-tobermorite and potassium bromide. Furthermore, the narrow band near 975 cm^−1^ is assigned to Si-O stretching vibration in silica tetrahedron. This band exhibited a slight shift to lower frequency continuously with the increasing of Al/(Si + Al) molar ratio. The intensity of this band was not affected by the increasing Al content. The band appearing at 455 cm^−1^ is assigned to bending vibration of Si-O in silica tetrahedron. This band also showed a slight shift to lower frequency but not continuously. It shifted to 448 cm^−1^ from 455 cm^−1^ when the Al/(Si + Al) molar ratio increased from 0 to 0.05 and then remained stable even though the Al/(Si + Al) molar ratio increase to 0.2. The shift of these two bands was caused by the substitution of Si^4+^ by Al^3+^ in the structure of tobermorite. However, the addition of Al had little effect on the intensity of these two bands. Xiaolu Qu reported a different results that the intensity of stretching vibration at 970 cm^−1^ significantly decreased in the presence of higher Al content [4]. This decrease of intensity was mostly caused by the decreased polymerization degree of the silicate chains. The stretching and bending vibration of Si-O band is sensitive to the change of Si/Al ratio in C-A-S-H network. The Al-O has a longer bond length and lower bond energy than that of Si-O, which makes a smaller elastic constants and a lower vibrational frequency of Al-O than those of Si-O. Thus, the vibrational frequency shifted to lower frequencies along with the increasing content of Al in the C-A-S-H network.

### 3.4. ^29^Si and ^27^Al MAS–NMR

The ^29^Si MAS–NMR spectra of Al-tobermorite with various Al/(Si + Al) molar ratios are presented in Figure 6. The relative content of each resonance peaks calculated from the fitted results are shown in Table 1.

The ^29^Si MAS–NMR spectra of tobermorite without Al substitution (Figure 6a) displayed three well-resolved signals at −78, −82, and −93 ppm, which means there are three different silicon sites in the structure of tobermorite. These three resonance peaks can be assigned to the silicon sites Q^1^, Q^2^, and Q^3^ in the structure of tobermorite, respectively. The relatively higher intensity of Q^2^ than Q^1^ and Q^3^ peaks indicates a longer chain length in the b axis direction. As shown in Figure 6b,c, a new signal at −88 ppm was detected while the intensity of Q^3^ was significantly decreased. The peak at −88 ppm can be assigned to the silicon non-bridging sites Q^3^(1Al). The intensity of Q^2^ sites did not change, while the intensity of peak at −78 ppm increased significantly. According to the SEM images (Figure 2d), the increment of the peak intensity at −78 ppm could be due to the smaller size of Al-tobermorite with Al/(Si + Al) = 0.2 which has a shorter silicate chains that contents more Si in Q^1^ sites. This increment could also be explained by the Al substitution of Si in bridging Q^2^ sites. The substituted site could be detected as Q^2^(1Al) sites and the resolved signals of Q^2^(1Al) sites is overlapped with Q^1^ sites.

This explanation is further confirmed by ^27^Al MAS–NMR, as shown in Figure 7. The assignment of ^27^Al MAS–NMR chemical shifts has been reported that the ^IV^Al resonates at 50–70 ppm should be present in the bridging position that links two silicate chains together, the ^V^Al resonates at 33–39 ppm is attributed to Al^3+^ substituting Ca^2+^ in the interlayer space of C-S-H, and the ^VI^Al resonates at 0–10 ppm coming from an amorphous aluminum hydroxide or a calcium aluminate hydrate at the C-S-H surface [12]. Only two resonating signals in the tetrahedral range, 53 ppm and 63 ppm, were detected in Al-tobermorite with Al/(Al + Si) = 0.1, which confirm all Al atoms are introduced in the silicate chains of tobermorite. The bridging sites at 63 ppm and non-bridging sites at 53 ppm are corresponding to Q^2^(1Al) and Q^3^(1Al), respectively. Furthermore, the relatively peak intensity of the Q^2^(1Al) sites at 63 ppm is higher than that of Q^3^(1Al) sites at 53 ppm, indicating more Si in Q^2^ sites than in Q^3^ sites was substituted by Al.

Figure 8 shows the structure of Al-substituted tobermorite with Al located in the bridging and non-bridging sites. The Al substitution for Si caused a regional charge imbalance which means there are electrostatic repulsive-forces in the substituted site. This electrostatic repulsive force makes the Al-O-Al unstable so that the neighboring Si tetrahedrons unavailable for Al substitution (as marked in Figure 8). Furthermore, two Si tetrahedrons on the other side of CaO layer are also restricted by this electrostatic repulsive-force (as marked in Figure 8). Thus, the Al substitution for Si in the structure of tobermorite is limited to about 15 at % which is calculated by the results of SEM-EDS. However, if Na^+^ is introduced in the reaction system, this substitution limit (15 at %) could be higher. Since the regional charge imbalance caused by Al substitution can be compensated by the absorbed Na^+^ [12].

### 3.5. Adsorption Capacity

In order to investigate the effect of Al substitution on the cation adsorption capacity of tobermorite, tobermorite, and Al-substituted tobermorite with Al/(Al + Si) = 0.15 were chosen for the adsorption capacity test (as described in Section 2.3), and the results are shown in Table 2.

The concentration of NaOH solution used for the test is 0.55 mol/L. It decreased to 0.43 mol/L and 0.38 mol/L after 10 mins of contacting with tobermorite and Al-substituted tobermorite with Al/(Al + Si) = of 0.15, respectively. The calculated adsorption capacity of Al-substituted tobermorite with Al/(Al + Si) = 0.15 is 1.37 mmol/g, which is about 44% higher than that of tobermorite without Al substitution. The adsorption of tobermorite to Na^+^ is physical adsorption which is closely related to the specific surface area. Thus, the improved adsorption capacity of Al-substituted tobermorite to Na^+^ is mostly due to the larger specific surface area caused by the loosened structure on z-axis.

## 4. Conclusions

Plate-like tobermorite and Al-substituted tobermorite with varies Al/(Si + Al) molar ratio have been prepared successfully via static hydrothermal synthesis at 180 °C for 15 h. The introduction of Al promoted the polymerization degree of the silicate chains of tobermorite. The interplanar crystal spacing of (002) increased after Al substitution due to the larger radius of Al^3+^ than Si^4+^. The Al substitution makes the structure of tobermorite become looser on the z axis direction. The maximum Al substitution ratio is limited to 15 at %, which is due to the regional charge imbalance after Si was substituted by Al. The weak endothermic peak at 680 °C and sharp exothermic peak at 840 °C in the DSC curve were assigned to the breaking of Si-OH bond and the recrystallization into β-wollastonite respectively. The larger exothermic peak area at 850 °C of tobermorite with Al/(Al + Si) molar ratio of 0.15 indicating that the Al-O-Si bond was formed in the structure of tobermorite and the Al participated in the recrystallization. New chemical environments of Q^2^(1Al) sites and Q^3^(1Al) sites of Si have been detected after Al substitution. The introduced Al was presented in the structure of tobermorite as tetra-coordinated. The adsorption capacity was improved by 44% for Al-substituted tobermorite with Al/(Si + Al) molar ratio about 0.15.

## Figures and Tables

**Figure 1 materials-12-03765-f001:**
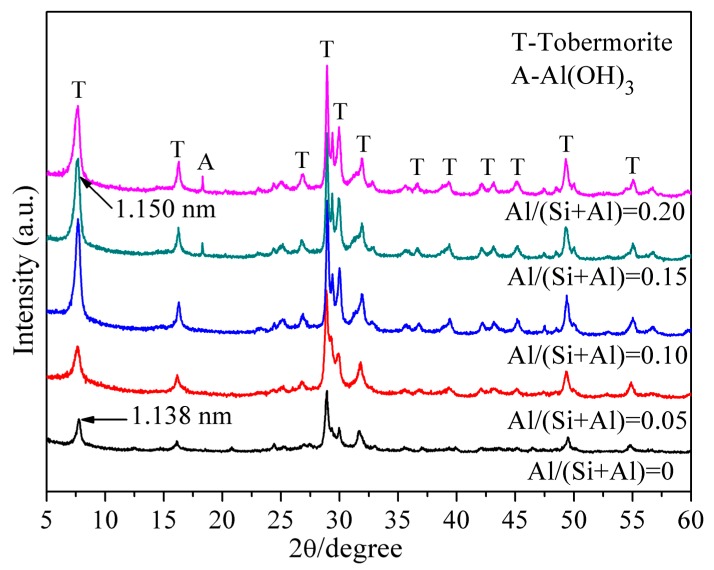
XRD patterns of hydrothermal synthesized Al-tobermorite with different Al/(Al + Si) molar ratios.

**Figure 2 materials-12-03765-f002:**
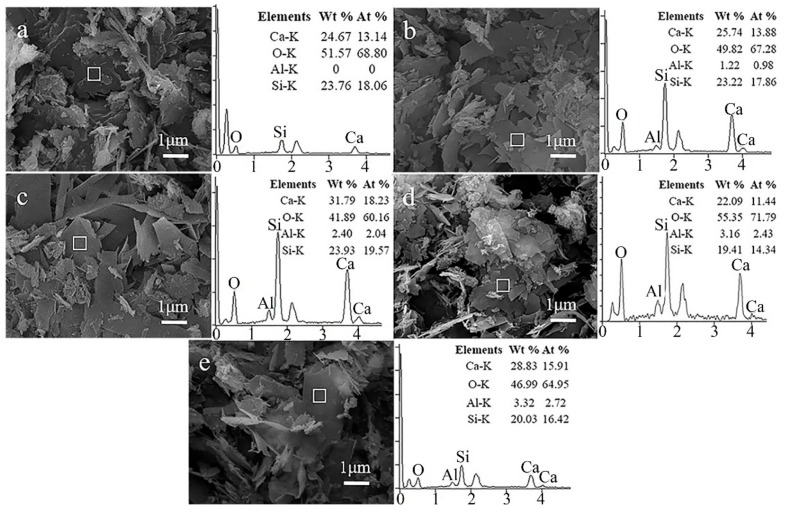
SEM–EDS images of hydrothermal synthesized Al-tobermorite with various Al/(Al + Si) molar ratios. ((**a**): Al/(Al + Si) = 0, (**b**): Al/(Al + Si) = 0.05, (**c**): Al/(Al + Si) = 0.1, (**d**): Al/(Al + Si) = 0.15, and (**e**): Al/(Al + Si) = 0.2).

**Figure 3 materials-12-03765-f003:**
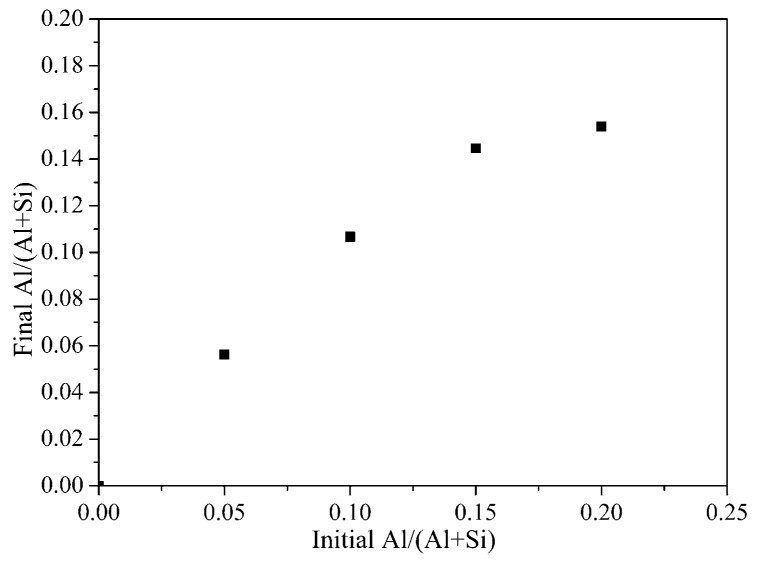
Correlation between initial Al/(Al + Si) molar ratio in reaction slurries and averaged final Al/(Al + Si) molar ratio detected by EDS.

**Figure 4 materials-12-03765-f004:**
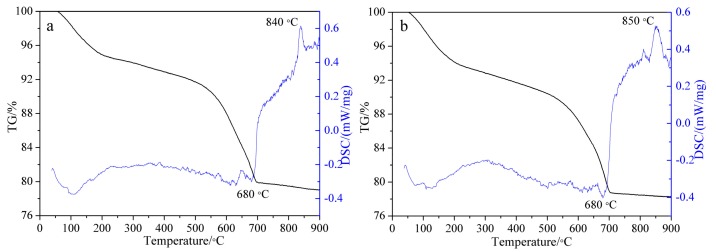
TG–DSC spectra of tobermorite and Al-substituted tobermorite with Al/(Al + Si) molar ratio of 0.15. (**a**): Al/(Al + Si) = 0, (**b**): Al/(Al + Si) = 0.15.

**Figure 5 materials-12-03765-f005:**
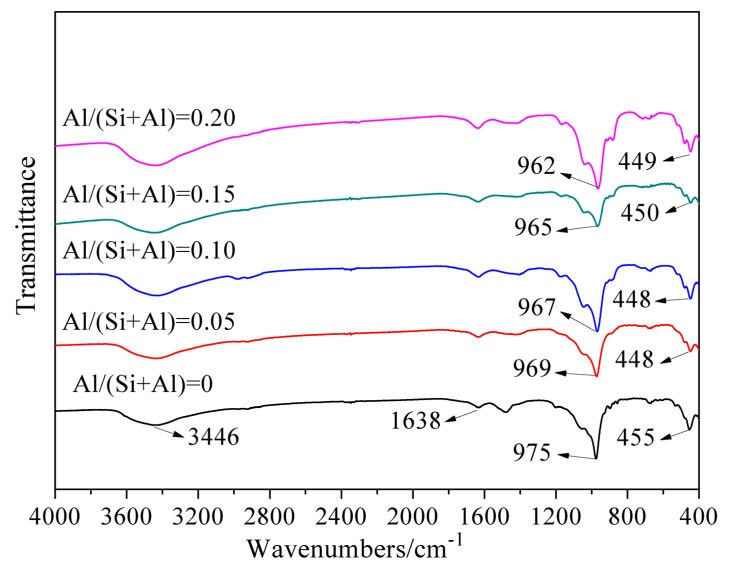
FT–IR spectra of Al-substituted tobermorite with different Al/(Si + Al) molar ratios.

**Figure 6 materials-12-03765-f006:**
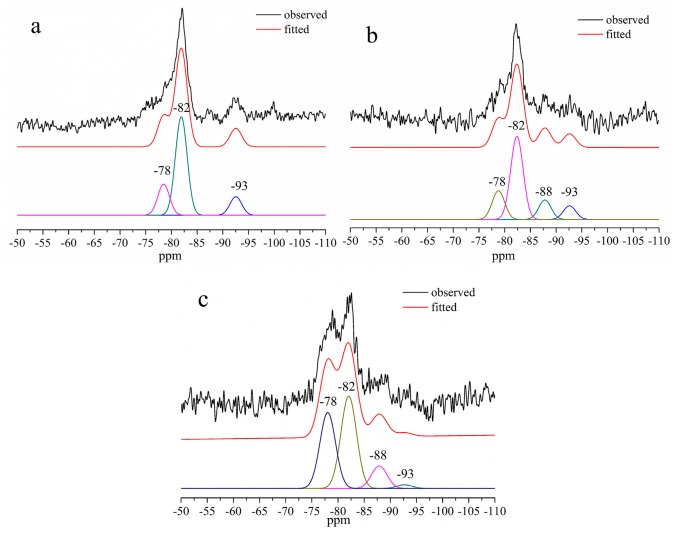
^29^Si MAS–NMR spectra of Al-tobermorite of various Al/(Al + Si) molar ratios. (**a**): Al/(Al + Si) = 0, (**b**):Al/(Al + Si) = 0.1, and (**c**):Al/(Al + Si) = 0.2.

**Figure 7 materials-12-03765-f007:**
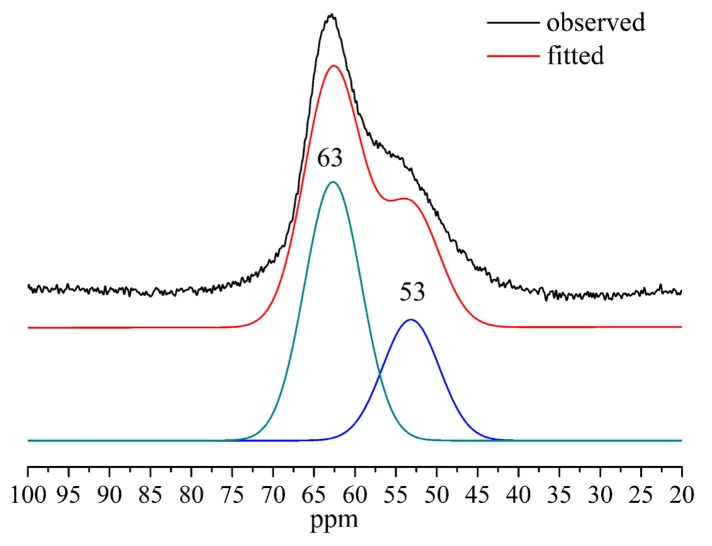
^27^Al MAS–NMR spectra of Al-tobermorite of Al/(Al + Si) = 0.1.

**Figure 8 materials-12-03765-f008:**
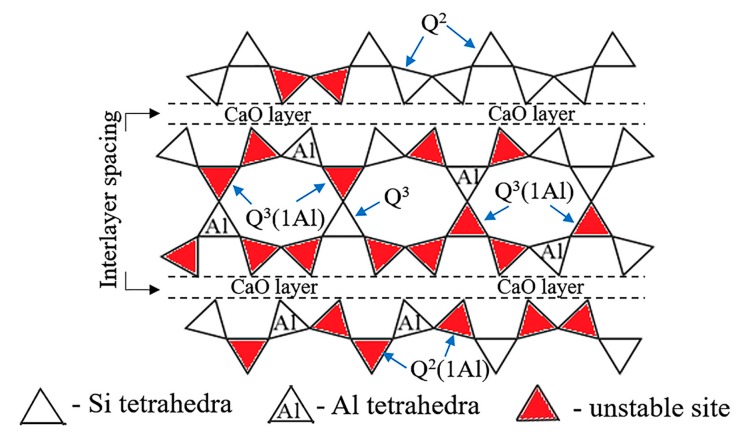
Schematic diagram showing unstable substitution sites in the structure of Al-substituted tobermorite.

**Table 1 materials-12-03765-t001:** Relative content of each resonance peak in ^29^Si MAS–NMR.

ppm	−78	−82	−88	−93
**Si Sites**	**Q^1^, Q^2^(1Al)**	**Q^2^**	**Q^3^(1Al)**	**Q^3^**
Al/(Al + Si) = 0	20.96%	66.56%	-	12.48%
Al/(Al + Si) = 0.1	19.85%	57.29%	13.44%	9.43%
Al/(Al + Si) = 0.2	39.03%	47.47%	11.60%	1.90%

**Table 2 materials-12-03765-t002:** Adsorption capacity of specimens to Na^+^.

-	Mass of Samples (g)	Volume of NaOH Solution (μL)	R(Na^+^) (mg/L)	c(Na^+^) (mol/L)	Adsorption Capacity (mmol/g)
NaOH solution	-	1200	8.5	0.55	-
Al/(Al + Si) = 0	0.1509	1200	6.7	0.43	0.95
Al/(Al + Si) = 0.15	0.1504	1200	5.9	0.38	1.37

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
