# Peer review of "Effect of Different Aluminum Substitution Rates on the Structure of Tobermorite"

_materials, 2019, doi:10.3390/ma12223765_

Round 1
Reviewer 1 Report
In the section describing the FT-IR results, replace the word "banding" with "bonding"
One never starts a sentence with words like "And" or "But", so please correct this.
Please ask a native English speaker to just read through this and correct the few places where the grammar of your sentences can be improved.
Reviewer 2 Report
General Comments:
The paper presents well-structured study on the Al-substituted tobermorite. In terms of originality the manuscript suits in the general trend of seeking the new cations exchangers that have selectivity properties, so, a content have a sufficient interest for the space required. XRD, SEM-EDS, FT-IR, 29Si and 27Al MAS-NMR, thus suitable for the Journal. Unfortunately, one might get an impression, that authors focus on the both preparation and editing of the manuscript as well as the formulation of the presented statements is insufficient. For this reason, some questions concerning this manuscript should be answered by the authors for improving the paper.
In particular, synthesis products must be investigated by simultaneous thermal analysis method, since in such a system often forms a semi-crystalline C-S-H(I) which is difficult or impossible to identify by other methods.
The work cites too much sources that were published 15 and more years ago. The introduction needs to be reviewed.
Page-line
1-32 Tobermorite should be labeled with nanometers (1.13 nm tobermorite), not angstroms.
2-60 A fineness of reagents used in the work must be reported. What modification of aluminum hydroxide was used?
3-96 Since the crystallinity of “pure” 1.13 nm tobermorite is low, I think that a mixture of this compound with C-S-H(I) forms. An STA is required.
Fig. 1 Surprisingly, even in the mixtures with high level of aluminum hydroxide did not form hydrogarnets. The STA would also help answer this question.
Fig. 2 How could you explain a significant increase of calcium in Figure 2 (c)?
Fig. 4. Poor quality of FT-IR curves. They are just described; there is no discussion with the results obtained by other authors.
8-206 In order to reliably prove that the cation adsorption capacity of Al-substituted tobermorite is higher, one test is not enough. I recommend investigating the sorption process under dynamic conditions.
I think that after a major revision including the suggested changes the paper could be accepted for publication in Journal.
Reviewer 3 Report
this paper is interesting and containing some knowledge, Al substitution seems interesting topic and effect of adsorption capacity is estimated. I have carefully checked and after following correction, it can be published.
Figure 2, word and number is too small to see and resolution is not so good.
Figure 4, figure and number seems bit small, resolution should be bit high.
Reviewer 4 Report
In this paper, the author have presented a “Effect of different aluminum substitution rates on the structure of tobermorite”. The paper is not suitable for publication:
The primary reason is that Authors have just varied the Aluminum doses in the tobermorite system, nearly identical studies with the similar experimental programme, findings and conclusions are already available.“Microstructure and characterization of hydrothermal synthesis of Al-substituted tobermorite” by Guo et al.
“Microstructure and characterization of aluminum-incorporated calcium silicate hydrates (C–S–H) under hydrothermal conditions” by Qu et al.
Round 2
Reviewer 2 Report
Many of my comments were taken into account by the authors. One essential point remains – the authors have not demonstrated the formation of "pure" 1.13 nm tobermorite (no endothermic effect at ~240 °C; exothermic effect at ~840 °C is typical for C-S-H (I) transition to wolastonite). I suggest that the authors would use the word "pure" to indicate that the product does not contain incorporated Al ions and mentions that they are likely to have synthesized a mixture of 1.13 tobermorite and C-S-H (I).
The authors changed the angstroms into nm in the text – the same must be done in Figure 1.
Reviewer 4 Report
In this paper, the author have presented a “Effect of different aluminum substitution rates on the 3 structure of tobermorite”. The paper needs major revision:
It is still not convincing that the current study has provided some new insights. The primary difference between the study by authors and below mentioned studies should be highlighted in detail along with details of how the findings of the present study are beneficial for a better understanding of tobermorite system with aluminum content .“Microstructure and characterization of hydrothermal synthesis of Al-substituted tobermorite” by Guo et al.
“Microstructure and characterization of aluminum-incorporated calcium silicate hydrates (C–S–H) under hydrothermal conditions” by Qu et al.
The language along with grammatical points of the manuscript should be checked. E.g L.48-49, 115-116 etc. Use of terms like “one year later” should be avoided in literature review. L.30-31.Author Response
Please see the attachment.
